# Changes in addressing inequalities in access to hospital care in Andhra Pradesh and Maharashtra states of India: a difference-in-differences study using repeated cross-sectional surveys

Mala Rao,[1,2] Anuradha Katyal,[3] Prabal V Singh,[3] Amit Samarth,[4] Sofi Bergkvist,[3] Manjusha Kancharla,[5] Adam Wagstaff,[6] Gopalakrishnan Netuveli,[7] Adrian Renton[1]

For numbered affiliations see end of article.

**Correspondence to**
Dr Mala Rao;
m.rao@uel.ac.uk

## ABSTRACT

**Objectives:** To compare the effects of the Rajiv Aarogyasri Health Insurance Scheme of Andhra Pradesh (AP) with health financing innovations including the Rashtriya Swasthya Bima Yojana (RSBY) in Maharashtra (MH) over time on access to and out-of-pocket expenditure (OOPE) on hospital inpatient care.

**Study design:** A difference-in-differences (DID) study using repeated cross-sectional surveys with parallel control.

**Setting:** National Sample Survey Organisation of India (NSSO) urban and rural 'first stratum units', 863 in AP and 1008 in MH.

**Methods:** We used two cross-sectional surveys: as a baseline, the data from the NSSO 2004 survey collected before the Aarogyasri and RSBY schemes were launched; and as postintervention, a survey using the same methodology conducted in 2012.

**Participants:** 8623 households in AP and 10 073 in MH.

**Main outcome measures:** Average OOPE, large OOPE and large borrowing per household per year for inpatient care, hospitalisation rate per 1000 population per year.

**Results:** Average expenditure, large expenditures and large borrowings on inpatient care had increased in MH and AP, but the increase was smaller in AP across these three measures. DIDs for average expenditure and large borrowings were significant and in favour of AP for the rural and the poorest households. Hospitalisation rates also increased in both states but more so in AP, although the DID was not significant and the subgroup analysis presented a mixed picture.

**Conclusions:** Health innovations in AP had a greater beneficial effect on inpatient care-related expenditures than innovations in MH. The Aarogyasri scheme is likely to have contributed to these impacts in AP, at least in part. However, OOPE increased in both states over time. Schemes such as the Aarogyasri and RSBY may result in some positive outcomes, but additional interventions may be required to improve access to care for the most vulnerable sections of the population.

## Strengths and limitations of the study

- This study uses two cross-sectional surveys to compare changes between the Indian states of Andhra Pradesh and Maharashtra, in hospital inpatient care-related expenditures and behaviours before and after the roll-out of the Aarogyasri and Rashtriya Swasthya Bima Yojana schemes.
- The study, based on a survey of 18 696 households, has shown that health innovations in Andhra Pradesh had a greater beneficial effect on hospital inpatient care-related expenditures and access than innovations in Maharashtra. The Aarogyasri scheme is likely to have contributed to these impacts in Andhra Pradesh.
- The study also highlights the implications of the findings for policy and practice, and additional interventions are necessary to address gaps in the availability of and access to care.
- The study is only able to compare the effects of health innovations over time across the two states but does not allow the drawing of inferences on the impacts of individual initiatives.
- The study uses the difference in differences methodology and its findings may have been affected by unobservable differential changes between the two states.

## INTRODUCTION

In 2005, member states of the WHO committed themselves to developing their health financing systems to deliver universal coverage (UC), so that all people have access to health services and do not suffer financial hardship when paying for them.[1] WHO's 2010 World Health Report[2] recommended inter alia that countries reduce reliance on direct payments, and improve equity of access, including through the introduction of prepayment schemes. However, one recent

systematic review of the impact of national health insurance schemes in low-income and middle-income countries[3] found only weak evidence of increased use of healthcare and reduced out-of-pocket expenses, with the poorer benefiting less, while others[4][5] concluded that health insurance improved healthcare access and use, as well as financial protection in most cases, but had no conclusive impact on health status. Both highlighted the need for more rigorous assessments of such schemes.

India has one of the highest levels of out-of-pocket health expenditure (more than 80% of private health expenditure).[6] The aim of our study was to explore how recently introduced health financing initiatives have affected access to and out-of-pocket expenditure (OOPE) on inpatient hospital care in the Indian states of Andhra Pradesh (AP) and Maharashtra (MH).

## BACKGROUND
In its *Eleventh Five Year Plan (2007–2012)*,[7] the Government of India (GOI) sought to increase public expenditure on health and to strengthen investment in the rural health infrastructure through the National Rural Health Mission. Its *Twelfth Five Year Plan (2012–2017)*[8] reflected the recommendation of the Planning Commission's High Level Expert Group for general taxation to be the principal source of healthcare financing. It proposed the development of government-funded health insurance schemes, building on the evidence from experimental schemes being introduced across many states. In India, health is primarily a state rather than national responsibility.[6] While it is recognised that political will and good governance are essential, both the political mobilisation of funds for health schemes and the effectiveness and efficiency of funded schemes will be enhanced by robust evidence which documents as to whether schemes achieve objectives, what works well and the main challenges they face.

In India, evaluation is not routine, even of large, costly public healthcare programmes. Nevertheless, some assessments have been carried out, for example, of the Yeshasvini Co-operative Farmers Health Care Scheme of Karnataka, the longest running state-supported health insurance scheme for the informal sector in India,[9] with promising results in terms of increased utilisation of and reduced borrowing for healthcare services.[10] However, these early models covered small populations and offered limited benefits, so that the policy implications of conclusions drawn from even the best evaluations were unclear.

This scenario has changed in the past 5 years with the launch of two schemes, the Rajiv Aarogyasri[11] Community Health Insurance Scheme (Aarogyasri) of AP and Rashtriya Swasthya Bima Yojana (RSBY) currently offered in 30 states and union territories of India,[12] including AP's neighbouring state of MH. Their scale in terms of population coverage and range of treatments offered significantly enhances their potential to inform India's road map towards universal health coverage. Both schemes belong to the new generation of publicly funded government-sponsored health insurance schemes, principally aimed at providing financial protection to the poor against catastrophic health shocks, which, for these schemes, the Government has defined as inpatient hospital care.[9] Both schemes have been subject to some assessment in their early phases, but a recent editorial[13] highlighted the necessity of further and repeated evaluations to enhance the credibility and accountability of existing schemes and to identify those which deserve a scale-up. The Aarogyasri and RSBY schemes and the other recent health sector innovations in AP and MH, which form the major backdrop and complex healthcare architecture against which these have been launched, and which may have contributed to the changes in outcomes we have explored in our study, are described below.

### The Rajiv Aarogyasri Health Insurance Scheme of AP
In 2007, AP launched a pioneering new state-wide, fully state-funded health insurance scheme, the Rajiv Aarogyasri Community Health Insurance Scheme (Aarogyasri),[11] to provide treatment for serious and life-threatening illnesses. The specific objectives include: to improve access of poor families to quality medical care (meaning low-frequency, high-cost specialist care) and treatment of identified diseases requiring hospitalisation through an identified network of healthcare providers, to provide financial cover for catastrophic illnesses which have the potential to wipe out the lifetime savings of poor families and to provide UC to the urban and the rural poor in the state,[14] albeit for the conditions covered in the benefits package. All families with a 'below poverty line' (BPL) ration card, that is, those on an annual income below US$1384 (₹75 000) in urban areas and US$1107 (₹60 000) in rural areas, and including individuals with pre-existing medical conditions are *automatically* enrolled and the scheme was estimated to cover approximately 20.4 million poor and lower middle class families, comprising about 85% of the state's population in 2009.[9] Enrollees make no contribution, the annual benefit is a maximum of US$4500 (₹200 000) per family per year and there is no limit on the size of the family.[14] A total of 942 medical and surgical procedures across 31 clinical specialties[14] are provided and the benefits include all inpatient costs—associated investigations, food, transport and medicines for 10 days following discharge. One year follow-up packages including consultation, medicines and diagnostics are also available for 125 procedures requiring longer periods of follow-up.[9] Aarogyasri has unique features including *Aarogyamithras* (health system navigators), outreach *health camps* delivered by participating hospitals to educate, screen and case-find and a state-of-the-art information technology-based management system. At the time of this study, 353 public and private sector hospitals

were 'empanelled' to provide services to Aarogyasri beneficiaries.

In 2009, a descriptive study of *Aarogyasri*, based on an analysis of claims data and a survey of beneficiaries,[15] concluded that while the scheme was beginning to reach its intended beneficiaries, uptake was lower among scheduled castes and tribes. This was confirmed by Fan *et al*[16] who used variations in programme roll-out over time and districts to evaluate the scheme using National Sample Survey data collected before and after its launch. They reported reduced OOPE in this initial phase but no major impact on catastrophic healthcare expenditure. Inspired by Aarogyasri and mindful of the political benefits of introducing popular health reforms, other states have launched health financing innovations similar to this model.

## RSBY in MH

RSBY was launched across a number of states by the Ministry of Labour, GOI in 2008[17] and provides access to free inpatient hospital care up to US$550 (₹30 000) per family per year.[18] Households which meet the criteria based on the much more limiting definition of poverty and numbers of poor families provided for each State by the GOI Planning Commission are eligible to enrol, and pay a contribution of US$0.55 (₹30) at registration and at each annual renewal.[9] Up to five family members, including those with pre-existing conditions, can be covered, and personal information including biometric data are collected prior to the issue of a smart card with encoded details of the family. Seven hundred procedures covering 18 broad categories of interventions, which would generally be included under the umbrella of 'secondary' care, are provided and the benefit packages include the intervention, public transport costs limited to US$1.8 (₹100) per visit and US$18.2 (₹1000) per year and posthospitalisation drugs for 5 days. Networked hospitals are required to provide free outpatient consultations (which have only recently been introduced; Kurian OC, personal communication), but other costs such as ambulatory diagnostics and medicines have to be borne by the beneficiaries, except if investigations lead to inpatient admissions within a day.[9] A pilot of the RSBY scheme was launched in one district of AP, but only after the start of our household survey.

In MH, enrolment began in August 2009, and by mid-2013, approximately two million of the eligible four million families were enrolled in the scheme,[12] which is being implemented in 31 of 35 districts in the state. Enrolment had extended to 26 districts prior to June 2012, when our household survey began. Notably, only 15 of 1215 hospitals contracted for RSBY funded services are from the public sector.[12] Early assessments of the scheme's impact nationally suggest that although the rate of hospitalisations has increased, awareness of the scheme was poor and remains a barrier to uptake. Notable variations in enrolment and scheme awareness were also observed by a descriptive study of RSBY conducted in the Amravati district of MH, which has a large tribal population.[19] In MH,[17] utilisation rates have been reported to be lower than in other states and the male:female enrolment ratio is 6.5:3.5.

## Other major health sector initiatives in AP and MH

Both states have a complex healthcare landscape with numerous programmes in place. There are several initiatives which have been launched during the past decade, some of which are common to both states and driven by national strategies, and others which owe their existence to state-level enterprise, innovation and political support. The most notable programmes with the potential to impact on patient care are described below.

The National Rural Health Mission (NRHM) was launched in 2005 nationwide, with the key aim of reducing maternal and infant mortality.[20] Government reports suggest that its notable achievements include an increase in institutional deliveries; in AP from 1.25 million in 2005–2006 to 1.46 million by 2011–2012[21] and in MH from 1.1 million to 1.63 million,[22] achieving an institutional delivery rate of approximately 92% in both states. Also common to both states is the '104 health information help line' launched in AP in 2008 and in MH in 2011[23] to provide medical advice and information based on validated algorithms and disease summaries, to direct callers to appropriate health facilities or to receive complaints against a public sector health facility. In AP, the help line and call centre were subsumed within the Aarogyasri infrastructure by 2011.

In MH, the RSBY was preceded by the Jeevandayee scheme launched in 1997 with the objective of reducing catastrophic OOPE on inpatient care in the BPL population.[24] Potential beneficiaries were required to apply for funding after a diagnosis was confirmed and the scheme covered serious illness such as cardiac and renal disease and cancer. However, the scheme uptake has been low, and while it has continued to run in parallel to the RSBY, only 66 853 procedures (4456 procedures per year in a state with 112.37 million people) have been approved during the scheme's lifetime.[25] Since 2006, MH has also had a scheme in place which mandated 20% of the beds in private hospitals to be made available for free or at subsidised rates to poor patients (Kurian OC). It has been estimated that around 10 000 private beds are available for the poor across MH, equivalent to approximately 20% of the total bed capacity of the public sector. Although the implementation of the scheme is reported to be erratic (Kurian OC), it may have had some positive impact on access to hospital inpatient care for serious illness.

Launched in 1995–1996, the Navasanjeevani Yojana scheme is exclusive to the 15 tribal districts of MH and was to improve maternal and infant mortality in these vulnerable populations.[26] It has focused on strengthening primary health and nutrition services and access to safe drinking water.

A service available in AP but not in MH is the '108' scheme, launched in 2005 to provide a state-of-the-art

medical emergency response service.[27] At the time of our study, 802 ambulances catered to approximately 3500 emergencies per day.[28]

## OBJECTIVES OF THE STUDY
Our objective was to compare the effects of health innovations over time on access to and OOPE on inpatient care in AP and MH and to assess whether the AP initiatives had larger or smaller beneficial effects than those found in MH. These differential effects are likely to be substantially due to the Aarogyasri scheme in AP and the RSBY in MH. In this paper, we report findings from a study which compared these changes. The findings do not allow us to draw inferences on the impacts of individual initiatives, but nevertheless contribute new knowledge on the impact and role of the innovations, provide lessons for other programmes, and strengthen the evidence base for policy on UC in India.

## METHODS
### Overview
None of the aforementioned initiatives—including the Aarogyasri and RSBY schemes in which we are especially interested—was piloted in a systematic way, let alone via a carefully designed randomised control trial. Following Medical Research Council (MRC) Guidance[29 30] and best practice, we therefore opted for a cross-sectional survey design in which we seek to minimise selection bias, to control for confounding variables and to reduce the effects of chance. Specifically, we compare changes in hospital inpatient care-related expenditures and behaviours (HREB) in AP and MH before and after the roll-out of the Aarogyasri scheme in AP and the RSBY scheme in MH. The difference in changes between AP and MH is not an estimate of a specific initiative. Rather, it tells us whether, on balance, the AP initiatives have had larger (or smaller) beneficial effects than the MH initiatives, and if so how much more (or less) beneficial they have been. Since the NRHM was common to both states and the MH-specific initiatives were quite small in scale or unlikely to affect HREBs, any difference in change between the two states is quite likely to be mainly due to the differential effects of the Aarogyasri and RSBY programmes.

AP and MH have a broadly similar development profile as shown by the data below. AP's other socio-economically similar neighbouring states of Karnataka and Tamil Nadu had already introduced Aarogyasri-like schemes, and Odisha and Chattisgarh, the only other neighbours, had comparatively higher levels of socio-economic deprivation. HREBs were measured in AP and MH by two waves of household survey before (2004) and after (2012) the introduction of Aarogyasri and RSBY.

### Survey design
#### Baseline survey: 2004
We used the original data from the NSSO 60th decennial round household survey undertaken in 2004[32] to estimate baseline HREB estimates for AP and MH (table 1). This was the most recent round measuring morbidity profiles, use of healthcare services including hospitalised and non-hospitalised treatments and expenditures incurred. The household survey used a multistage stratified sampling methodology to identify a representative random population sample and an interviewer completed questionnaire to obtain measures of HREB along with sociodemographic household expenditure and other information.

#### Follow-up survey: 2012
We used the same household survey design and methods to collect postintervention data in AP and MH as those used by NSSO. Briefly, the household survey used a multistage stratified sampling methodology with the 'First Stage Units' (FSUs) identical to those used by NSSO in their 66th round (2008–2009),[33] the latest round for which FSUs had been mapped. However, the FSUs were not the same as those in NSSO 2004, our baseline survey, rapid urbanisation having changed substantially the urban–rural landscape of both states and thus the geographical basis for sampling units.

The interviewer-completed household survey questionnaire was pretested and then piloted in both states prior to the survey. All respondents provided written informed consent for participation. Questions addressed the following: household composition and sociodemographic characteristics of members, household expenditure, health expenditure (outpatient and inpatient) and means of its financing, healthcare-seeking behaviour, factors

| MH | Indicator | AP |
|---|---|---|
| 112.37 | Population (2011 census, in millions) | 84.66 |
| 10.20 | % Scheduled Caste (2001 census)* | 16.60 |
| 8.90 | % Scheduled Tribe (2001 census)* | 6.20 |
| 101 314 | Per capita income 2011–2012 (in ₹)† | 71 540 |
| 35 | Number of districts | 23 |
| 5314 | Households covered in National Sample Survey Organisation (NSSO) 60th round (2004–2005) | 5059 |
| 10 073 | Households covered in the study 2012 | 8623 |

*Note that the 2011 census data for social groups are not yet published.
†Source: Presentation on Annual Plan 2012–2013 and Five Year Plan 2012–2017.[31]

**Table 1** Urban and rural populations and households surveyed in 2004 and 2012 in Andhra Pradesh and Maharashtra

| | Andhra Pradesh | | Maharashtra | |
| --- | --- | --- | --- | --- |
| | NSSO 60th round 2004–2005 | Our survey 2012 | NSSO 60th round 2004–2005 | Our survey 2012 |
| Population | 76 210 007* | 84 665 533† | 96 878 627* | 112 372 972† |
| Urban population | 20 808 940* | 28 353 745† | 41 100 980* | 50 827 531† |
| Rural population | 55 401 067* | 56 311 788† | 55 777 647* | 61 545 441† |
| Total households (urban) | 4 397 138* | 6 778 225† | 8 403 224* | 10 813 928† |
| Total households (rural) | 12 607 167* | 14 246 309† | 11 173 512* | 13 016 652† |
| Total households | 17 004 305* | 21 024 534† | 19 576 736* | 23 830 580† |
| FSUs (urban) | 183 | 372 | 267 | 504 |
| FSU (rural) | 325 | 491 | 265 | 504 |
| Total households surveyed (urban) | 1824 | 3715 | 2664 | 5038 |
| Total households covered (rural) | 3235 | 4908 | 2650 | 5035 |

*2001 Census.
†2011 Census.
The NSSO 66th round had 492 rural FSUs in Andhra Pradesh, but 1 FSU was found to be uninhabited.
FSU, First Stratum Unit; NSSO, National Sample Survey Organisation of India.

affecting access to healthcare and awareness and perceptions of the quality of the *Aarogyasri* scheme (AP only). The survey questions in 2012 were identical to those from the NSSO 2004.[34] Additional questions specific to the Aarogyasri and other relevant schemes were also added.

A survey of 18 696 households across 2 states and 1871 locations within the states is a challenging undertaking. The survey design had several features intended to assure the quality of data collected. Few academic institutions have the internal capacity to carry out such large surveys, and consequently the Social and Research Institute of IMRB International, a leading market research agency, was selected to carry out the survey. The Institute has field survey teams based in every Indian state, conversant in local languages and dialects and trained to carry out surveys in the socioeconomic development sector. Its clients include the GOI (for whom the national Family Health Survey data are collected), the World Bank and other UN organisations. A group of NSSO consultants in AP and the Indian Socioeconomic Research Unit, Pune were recruited to support the training of the field survey teams and data verification.

We planned three levels of verification of the study data: the first to be undertaken by the survey agency, the second to be carried out by the study team and the third by the agencies mentioned above. Survey teams for each district were accountable to a field supervisor who was responsible for checking the household listing and data entry on a daily basis. The study team also accompanied the field staff to survey sites on a regular basis. Data collected from 250 households in each state (approximately 2.5% of the surveyed households) and 186 of the FSU listings (approximately 10%) were independently verified by the agencies in the villages and urban blocks in order to ensure that the sampling method and administration of the questionnaire survey were being correctly applied. The data entry was carried out by the Institute using a double entry method and any questionnaires reported incorrect were sent back to the field for resurvey. The research team carried out a final validation and review of the data.

## Outcome measures
### Average inpatient expenditure per household per year
Average OOPE for inpatient care during the 1 year prior to the survey was estimated from questionnaire responses for AP and MH from the baseline and follow-up data. Reimbursements for inpatient expenditure were deducted from the total where households had received them.

### Large out-of-pocket inpatient expenditure
Owing to the limited data on household consumption in the 2004 NSSO health survey, we did not estimate 'catastrophic health expenditure'. Instead, we constructed a measure of 'large' OOPE. The Aarogyasri Health Care Trust data on expenditure incurred by the Government of AP per case in 2012 were examined[14] and the mean was estimated as US$419 (₹23 000). A household was deemed to have incurred 'large' expenditure if OOPE for inpatient care was equal to or greater than this threshold.

### Large borrowing
We estimated the total amount borrowed by a household to meet the expenditure of all the inpatient episodes of that family during the previous year. A household was considered to have incurred a 'large borrowing' if the borrowing was equal to or exceeded the BPL threshold set by the Government of AP: ₹70 000 for urban families and ₹65 000 for rural households. These prices have been deflated to 2004 levels.

### Hospitalisation rate
This was estimated as the number of individuals hospitalised during the previous year, per 1000 population.

Variations in outcomes were examined between male-headed and female-headed households, and rural and urban populations, as well as across social groups and economic groups represented by asset quintiles.

Owing to the limited data on household consumption in the 2004 NSSO health survey which made the estimation of wealth difficult, we have opted instead to measure household living standards using an asset or wealth index based on information on ownership of household durables, dwelling type, etc, using principal component analysis to estimate weights[35–37] for each indicator. The indicators used were limited to those collected in the 2004 NSSO: type of structure of the dwelling unit, type of toilet, type of fuel used for cooking and source of drinking. Data from the 2004 and 2012 surveys were pooled so that the index captures changes in living standards between the 2 years; there are therefore more households in the top quintile in 2012 than in 2004. The statistical software Stata V.11 was used to generate the index.

### Deflation of follow-up expenditure estimates

The 2012 expenditure data including the threshold for large expenditures were deflated using the consumer price index of the GOI[38] to reflect 2004 prices.

### Analysis

Our repeat cross-sectional surveys do not allow for estimation of within-individual household changes in outcomes over time. Our analysis therefore focused on estimating outcomes averaged across states, and in comparing changes in these over time between AP and MH. If we assume that outcome determinants other than Aarogyasri and RSBY remained stable in the two states over time or followed a parallel change, then a difference-in-differences (DID) analysis will uncover the net effect of Aarogyasri over and above RSBY.

The DID of outcome ($Y_{DD}$) is

$$(Y_{2012}^{AP} - Y_{2004}^{AP}) - (Y_{2012}^{MH} - Y_{2004}^{MH})$$

where the subscripts and superscripts for Y refer to the respective states and the years when the surveys were carried out. CIs were calculated from the SE $Y_{DD}$ and the p value for the null-hypothesis ($Y_{DD} = 0$) was tested using the Wald test as $t = Y_{DD}/SE_{Y_{DD}}$ with one degree of freedom. $Y_{DD}$ was estimated using ordinary least square regression:

$$y_{it} = \beta_0 + \beta_1 state_i + \beta_2 survey_t + \beta_3 (state \times survey)_{it}$$
$$+ \sum_{k=1}^{m} \beta_{3+k} covariate_k + \varepsilon$$

The basic DID results are obtained using the above regression with covariates excluded. The adjusted DID results are obtained using the above regression with m=9 covariates, namely the gender of the head of the household, a dummy variable capturing whether the household lives in a rural or urban location, three dummy variables capturing the household's social group (the lowest is the excluded category) and four asset quintile dummies (the bottom is the excluded category). In the regression, $y_{it}$ is the outcome, *state* is a dummy variable with 0 for MH and 1 for AP, and *survey* is a dummy variable with 0 for the 2004 survey and 1 for the 2012 survey. The coefficient for the interaction term, $\beta_3$, gives the DID estimate, $Y_{DD}$. Robust SEs of $Y_{DD}$ were calculated to account for clustering of households within FSUs using Stata survey commands. A positive value for $Y_{DD}$ suggested that the change in the outcome in AP was more than the change in MH and that the negative value would suggest the reverse.

An advantage of a regression based DID estimate is this ability to use covariates which can account for differential trajectories in the two states. In addition to this, we did subgroup analysis stratifying for different covariates. This is particularly relevant in the case of scheduled tribes whose proportion increased in MH in the follow-up survey.

Subgroups were not mutually adjusted for the analysis due to sample size restrictions in relation to some of them.

### Role of the funding sources

The external funding sources had no role in study design, data collection, analysis, interpretation or reporting or in submission decision.

## RESULTS

A total of 5314 and 5059 households from MH and AP were surveyed by the NSSO in 2004 (table 2). Our survey in 2012 included 10 073 (MH) and 8623 (AP) households.

### Changes in average in-patient expenditure

Table 3 (top panel) shows average baseline levels of inpatient expenditure. The table also shows the real terms change (deflated to 2004 prices) in these outcomes at follow-up and the DID estimate comparing AP with MH. DIDs for overall results are shown unadjusted, as well as adjusted for the effects of the covariates. Breakdowns by sex of the head of the household, social group, urban/rural location and asset quintiles are also shown.

Overall, average inpatient expenditure increased in real terms in the states between 2004 and 2012, but the increase was significantly greater in MH (unadjusted DID=−498.2 ₹, 95% CI −792.9 to −203.5, p=0.0009). The direction in terms of a greater increase in MH was evident across all subgroups of analysis except the richest asset quintile. However, the DIDs reached significance in male-headed households (DID=−513.7 ₹, 95% CI −843.9 to −183.4, p=0.0023), scheduled castes (DID= −708.7 ₹, 95% CI −1234.3 to −183.2, p=0.0082), all 'other' social groups (DID=−1110.46 ₹, 95% CI −1868

**Table 2** Sociodemographic characteristics of baseline and follow-up samples

| Subgroups | Number (%) of households 2004 | | Number (%) of households 2012 | |
|---|---|---|---|---|
| | Maharashtra | Andhra Pradesh | Maharashtra | Andhra Pradesh |
| All | 5314 | 5059 | 10 073 | 8623 |
| Head of household | | | | |
| Male | 4785 (90.0) | 4433 (87.6) | 8543 (84.8) | 7418 (86.0) |
| Female | 529 (10.0) | 626 (12.4) | 1530 (15.2) | 1205 (14.0) |
| Social group | | | | |
| Scheduled Tribes | 413 (7.8) | 296 (5.9) | 1364 (13.5) | 883 (10.2) |
| Scheduled Castes | 809 (15.2) | 974 (19.3) | 2235 (22.2) | 1797 (20.8) |
| Other excluded | 1644 (30.9) | 2317 (45.8) | 1899 (18.9) | 3419 (39.7) |
| All other groups | 2448 (46.1) | 1472 (29.1) | 4571 (45.4) | 2524 (29.3) |
| Location | | | | |
| Rural | 2650 (49.9) | 3235 (63.9) | 5035 (50.0) | 4908 (57.0) |
| Urban | 2664 (50.1) | 1824 (36.1) | 5038 (50.0) | 3715 (43.0) |
| Asset quintile | | | | |
| Lowest | 1260 (23.7) | 1594 (31.5) | 996 (9.9) | 826 (9.6) |
| Second | 1016 (19.1) | 1237 (24.5) | 1841 (18.2) | 1286 (14.9) |
| Third | 772 (14.5) | 753 (14.9) | 2228 (22.1) | 2121 (24.60) |
| Fourth | 857 (16.1) | 744 (14.7) | 2373 (23.6) | 3072 (35.6) |
| Fifth | 1408 (26.5) | 730 (14.4) | 2633 (26.1) | 1318 (15.3) |

to −352.9, p=0.0041), rural households (DID=−504 ₹, 95% CI −801.9 to −206.0, p=0.0009) and the poorest (DID=−1001.3 ₹, 95% CI −1751 to −251.7, p=0.0089) and middle asset quintiles (DID=−798.1 ₹, 95% CI −1362.9 to −233.3, p=0.0056).

## Large expenditures for inpatient care

Proportions of households incurring large expenditures showed an increase in both states (table 4), but the increase was smaller in AP for the sample as a whole as well as for all the groups except for the second asset quintile. The DID was strongly significant for the households overall (adjusted DID=−1.8, 95% CI −3 to −0.7, p=0.0009), but this was not observed for any of the subgroups of analysis.

## Changes in large borrowing for inpatient care

In both states, proportions of households incurring large borrowings to meet inpatient expenses increased from 2004 to 2012 (table 5). However, there was a consistent pattern of smaller increases in AP for the overall population, as well as all subgroups (except the richest asset quintile) with DIDs strongly significant for the overall population (adjusted DID=−4, 95% CI −6.6 to −1.4, p=0.0032), scheduled tribes (DID=−5.5, 95% CIs −9.3 to −1, p=0.0048), rural households (DID=−4.7, 95% CIs −7.3 to −2.1, p=0.0007) and all asset quintiles except the richest (the poorest asset quintile DID=−9.0, 95% CI −14.0 to −4.4, p=0.0002).

## Hospital utilisation for inpatient care

Overall, hospitalisation rates have increased in AP and MH (table 6), but more so in AP (5.6/1000 population vs 2.2), although the DID was not statistically significant. The subgroup analysis presented a mixed picture. For both male-headed and female-headed households, there was a greater increase in hospitalisation in AP, but this reached moderate statistical significance only for female-headed households (DID=27.6, 95% CI 1.1 to 54.1, p=0.0415). There is an increase in hospitalisations among scheduled tribes in MH and a reduction in AP (DID=−19.8, 95% CI −37.3 to −2.3, p=0.0272), but the opposite picture was seen among 'other excluded' groups with an increase in AP and a reduction in MH (DID=12.5, 95% CI 1.2 to 23.9 p=0.0309). In scheduled castes, hospitalisations had increased in both states but more so in AP, while in the 'other' group there was a small increase in MH and a small reduction in AP. In the poorest quintile, the increase in hospitalisation was significantly greater in MH (DID=−14.4, 95% CI −28 to −0.31, p=0.0451).

## LIMITATIONS OF THE STUDY

DID estimations aimed at assessing the impacts of interventions assume that both populations demonstrate similar characteristics prior to the start of the intervention, and that 'unobservables' follow a common trend; under such circumstances, any differences in changes observed over time between the two populations are attributable to the interventions.[39 40] Despite AP and MH having broadly similar socioeconomic profiles, as well as our DID analysis taking account of a number of covariates, there may have been other factors resulting in unobserved differential changes between the two populations to which the results of the DID analysis may be at least partially attributable.

A second limitation could arise from the impact of other public health programmes implemented during the period 2004–2012. The most significant of these is the National Rural Health Mission launched in 2005,

**Table 3** Change in average inpatient expenditure (in ₹) in Maharashtra and Andhra Pradesh between 2004 and 2012

| Subgroups | Baseline mean (95% CI) | | Change 2004:2012 mean (95% CI) | | DID | |
| --- | --- | --- | --- | --- | --- | --- |
| | Maharashtra | Andhra Pradesh | Maharashtra | Andhra Pradesh | Mean (95% CI) | p Value |
| Household inpatient expenditure | | | | | | |
| All | 1091.6 (978.4 to 1204.8) | 723.5 (527.5 to 919.5) | 942.8 (749.9 to 1135.6) | 444.55 (221.5 to 667.6) | −498.2 (−792.9 to -203.5) | 0.0009 |
| | | | | | Adjusted for covariates | |
| | | | | | −565.8 (862.9 to −268.6) | 0.0002 |
| Head of household | | | | | | |
| Male | 1132.9 (1015 to 1251) | 758 (419.7 to 1096.3) | 935 (727 to 1143.01) | 1074.9 (555.9 to 1593.8) | −513.7 (−843.9 to −183.4) | 0.0023 |
| Female | 757.1 (555.5 to 1014.6) | 341.1 (222.3 to 460.01) | 421.3 (164.7 to 678.0) | 589.9 (307.22 to 872.8) | −484.9 (−1075.6 to 105.9) | 0.1076 |
| Social group | | | | | | |
| Scheduled Tribes | 376.6 (231.7 to 521.6) | 432.7 (212.52 to 652.9) | 1153.1 (803.3 to 1502.9) | 675.2 (163.2 to 1187.2) | −477.9 (−1097.7 to 142) | 0.1307 |
| Scheduled | 696.7 (500.2 to 893.2) | 432.6 (305.6 to 559.4) | 1464.1 (1039.9 to 1888.4) | 755.4 (444.9 to 1065.9) | −708.7 (−1234.3 to −183.2) | 0.0082 |
| Castes | | | | | | |
| Other excluded | 1028.6 (838.4 to 1218.8) | 562.4 (463.7 to 662) | 928.9 (532.9 to 1324.9) | 767.9 (569.6 to 966.2) | −161 (−603.7 to 281.7) | 0.4758 |
| All other groups | 1424.5 (1222.3 to 1626.7) | 1306.2 (627.8 to 1984.6) | 734.9 (427.9 to 1041.7) | −375.6 (−1068.5 to 317.4) | −1110.46 (−1868 to −352.9) | 0.0041 |
| Location | | | | | | |
| Rural | 897.8 (768.1 to 1027.5) | 571.4 (496.2 to 646.6) | 1084.7(826.3 to 1343.1) | 580.7 (432.2 to 729.2) | −504 (−801.9 to −206.0) | 0.0009 |
| Urban | 1343.5 (1146.1 to 1540.9) | 1113.5 (466.2 to 1760.8) | 753.6 (458.7 to 1048.6) | 92.3 (−586.92 to 771.5) | −661.3 (−1401.5 to 78.864) | 0.0799 |
| Quintile | | | | | | |
| Poorest | 656.3 (498.0 to 814.6) | 391.5 (319 to 464.1) | 1692.5 (1053.3 to 2331.7) | 691.2 (298.9 to 1083.5) | −1001.3 (−1751 to −251.7) | 0.0089 |
| 2nd | 786.5 (583.5 to 989.5) | 443.3 (356.5 to 530.2) | 979.3 (599.4 to 1359.2) | 839.5 (465.7 to 1213.3) | −139.8 (−672.5 to 393) | 0.607 |
| Middle | 1062.7 (738.8 to 1386.1) | 862.1 (577.8 to 1146.5) | 1011.8 (550.2 to 1473.4) | 213.7 (−112.1 to 539.6) | −798.1 (−1362.9 to −233.3) | 0.0056 |
| 4th | 1241.7 (894.4 to 1589.1) | 1819 (337.5 to 3302.5) | 803.6 (328.7 to 1278.5) | −644.3 (−2128.3 to 839.7) | −1447.9 (−3005.2 to 109.5) | 0.0684 |
| Richest | 1818.6 (1505.5 to 2131.8) | 908.3 (682.1 to 1133.4) | 252.3 (−193.4 to 698.1) | 362.1 (15.3 to 708.9) | 109.7 (−454.80 to 674.3) | 0.7031 |

DID, difference in differences.

**Table 4**  Change in the proportion (%) of households incurring large health expenditures for inpatient care in Maharashtra and Andhra Pradesh between 2004 and 2012

| Subgroups | Baseline mean (95% CI) | | Change 2004:2012 mean (95% CI) | | DID | |
| | Maharashtra | Andhra Pradesh | Maharashtra | Andhra Pradesh | Mean (95% CI) | p Value |
|---|---|---|---|---|---|---|
| Large inpatient expenditure (₹23 000 deflated to 2004 figures) | | | | | | |
| All | 6.7 (6 to 7.3) | 3.4 (2.9 to 3.9) | 3.1 (2.1 to 4.1) | 2.2 (1.5 to 2.8) | −0.91 (−2.1 to 0.27) | 0.1302 |
| | | | | | Adjusted for covariates | |
| | | | | | −1.8 (−3 to −0.7) | 0.0009 |
| Head of household | | | | | | |
| Male | 6.8 (6.2 to 7.6) | 3.5 (3.1 to 4) | 3.1 (2.0 to 4.1) | 2.1 (1.5 to 2.9) | −0.8 (−2.1 to −0.4) | 0.1928 |
| Female | 5.0 (3.7 to 6.4) | 2.8 (1.9 to 3.7) | 3.9 (2 to 5.8) | 2 (6.6 to 3.4) | −1.8 (−4.2 to 0.50) | 0.1222 |
| Social group | | | | | | |
| Scheduled Tribes | 2.2 (1.3 to 3.1) | 1.4 (0.5 to 2.2) | 5.3 (3.5 to 7) | 3.5 (1.9 to 5.1) | −1.7 (−4.1 to 0.61) | 0.1478 |
| Scheduled Castes | 6.1 (4.6 to 7.7) | 2.1 (1.5 to 2.6) | 4.3 (2.3 to 6.3) | 3 (1.9 to 4.1) | −1.2 (−3.5 to 1.01) | 0.2785 |
| Other excluded | 5.9 (4.7 to 7.0) | 3.3 (2.7 to 3.8) | 2.9 (1.1 to 4.6) | 2.7 (1.7 to 3.6) | −2.1 (−2.2 to 1.8) | 0.8389 |
| All other groups | 8.3 (7.5 to 7.9) | 5.4 (4.4 to 6.3) | 2.2 (0.9 to 3.6) | 0.51 (−0.7 to 1.7) | −1.7 (−3.5 to 0.04) | 0.0628 |
| Location | | | | | | |
| Rural | 1.9 (1.5 to 2.2) | 0.9 (0.79 to 1.1) | 1.7 (1.1 to 2.3) | 1.3 (0.09 to 1.6) | −0.45 (−1.1 to 0.25) | 0.2098 |
| Urban | 12.9 (11.9 to 14) | 9.7 (8.9 to 10.7) | 4.4 (3.0 to 5.7) | 3.9 (2.6 to 5.3) | −0.7 (−2.4 to 1.5) | 0.6350 |
| Quintile | | | | | | |
| Poorest | 1.8 (1.3 to 2.4) | 1.1 (0.8 to 1.4) | 3.7 (2.2 to 5.2) | 1.7 (0.7 to 2.7) | −0.2 (−3.8 to −0.19) | 0.0307 |
| 2nd | 2.7 (1.9 to 3.5) | 1.2 (0.9 to 1.5) | 2.1 (0.93 to 3.3) | 2.2 (1.4 to 3.1) | 0.9 (−1.4 to 1.6) | 0.9079 |
| Middle | 6.9 (4.9 to 8.9) | 4.2 (3.1 to 5.3) | 1.3 (−1 to 3.6) | 0.9 (−1.2 to 1.4) | −1.2 (−3.9 to 1.4) | 0.3596 |
| 4th | 10.7 (8.7 to 12.6) | 7.6 (5.9 to 9.2) | 1.8 (−0.57 to 4.3) | −0.036 (−1.9 to 1.80) | −1.9 (−4.9 to 1.2) | 0.2268 |
| Richest | 13.5 (12 to 14.9) | 9.6 (8.1 to 11.2) | 0.3 (−1.6 to 2.2) | −0.6 (−2.8 to 1.6) | −0.9 (−3.7 to 2) | 0.5601 |

DID, difference in differences.

mainly to improve maternal and child health through the revitalisation of rural primary care and child and maternal health services. A key assumption of our study is that the impacts of the NRHM in terms of healthcare expenditure for maternal and child healthcare would have been similar in both states, as this was a nationwide development. Despite the improvements in the public sector maternal and child health services sought by the NRHM, it is widely recognised that the public, including BPL families, continues to pay OOP for private healthcare. We have assumed that this behaviour is likely to be similar across the two states. Other health initiatives, such as the Navsanjeevani Yojana and the 104 helpline, were unlikely to have had an impact on inpatient care or expenditure. The 108 scheme had the potential, in AP, to influence hospitalisation rates by helping more households to visit hospitals when seriously ill. However, the effect is likely to be small as the majority of even serious illnesses do not result in a 108 call for transport. The implementation of the RSBY and the scheme to make private hospital beds available for the poor in MH may have diluted the DID, although both schemes are known to have been only partially implemented.

Lastly, the 2004 NSSO survey, which served as our baseline, was carried out between January and June 2004. Our end-line 2012 survey was carried out over a period of 3 months from June to September. The morbidity and mortality patterns recorded in different time periods may vary, and could have influenced the data.

## DISCUSSION

We found that average expenditure, large expenditures and large borrowings on inpatient care had increased in MH and AP, but the increase was consistently smaller in AP across these three outcome measures, which may be suggestive of Aarogyasri having a somewhat larger effect than RSBY. Similar increases in institutional deliveries across the two states, as well as low levels of utilisation of RSBY and Jeevandayee schemes in MH, may further strengthen this explanation.

The increase in average OOPE on inpatient care in AP and MH reflects a pattern observed nationwide.[16 41] The Aarogyasri scheme may have contributed to the more favourable trajectory in AP directly and indirectly, in that the scheme may have contributed to a reduction in the prices of interventions and an increase in competition among healthcare providers. The evaluation of the Yeshasvini scheme also found a significant reduction in the price of surgical interventions.[10] Our findings may suggest that the positive effects of Aarogyasri detected by other studies[15 16] at an early stage of the roll-out of the scheme have been sustained. Automatic enrolment on the scheme, near universality of coverage and no requirement for enrollee contributions may have contributed to the significant DIDs in male-headed households, scheduled castes, rural households and the poorest and middle asset quintiles. However, these benefits were not demonstrated in some of the most vulnerable groups—female-headed households and scheduled

**Table 5** Change in the proportion (%) of households' large borrowings for inpatient care in Maharashtra and Andhra Pradesh between 2004 and 2012

| Subgroups | Baseline mean (95% CI) | | Change 2004:2012 mean (95% CI) | | DID | |
| --- | --- | --- | --- | --- | --- | --- |
| | Maharashtra | Andhra Pradesh | Maharashtra | Andhra Pradesh | Mean (95% CI) | p Value |
| Proportion of households having large borrowings | | | | | | |
| All | 7.5 (6.7 to 8.2) | 3.8 (3 to 4.5) | 8.9 (6.8 to 11) | 5.3 (3.4 to 7.2) | −3.7 (−6.4 to −0.908) | 0.0100 |
| | | | | | Adjusted for covariates | |
| | | | | | −4 (−6.6 to −1.4) | 0.0032 |
| Head of household | | | | | | |
| Male | 7.8 (7.0 to 8.5) | 3.9 (3.1 to 4.7) | 9.8 (6.6 to 1.3) | 5.3 (3.3 to 7.3) | −3.6 (−6.6 to −0.62) | 0.0187 |
| Female | 5 (3 to 7) | 2.9 (1.7 to 4.1) | 8.(6.6 to 11.2) | 5 (2.9 to 7.23) | −4.7 (−8.3 to −1) | 0.0137 |
| Social group | | | | | | |
| Scheduled Tribes | 3.6 (2.3 to 4.9) | 2.4 (0.93 to 3.9) | 11 (8.9 to 14) | 5.8 (2.8 to 8.8) | −5.5 (−9.3 to −1.8) | 0.0048 |
| Scheduled Castes | 7.2 (5.5 to 8.8) | 3 (1.7 to 4.2) | 9.6 (6.6 to 13) | 5.8 (3.4 to 8.3) | −3.8 ( −7.5 to 0.03) | 0.0518 |
| Other excluded | 8.0 (6.8 to 9.2) | 3.5 (2.7 to 4.4) | 8 (5.8 to 10.3) | 5.3 (3.2 to 7.4) | −2.8 (−5.7 to 0.19) | 0.0661 |
| All other groups | 8 (7.15 to 8.8) | 0.052 (.040 to 0.064) | 8.8 (5.9 to 12) | 4.7 (2.2 to 7.4) | −4.1 (−7.9 to −0.4.0) | 0.0302 |
| Location | | | | | | |
| Rural | 6.5 (5.6 to 7.5) | 0.03 (0.024 to 0038) | 10 (8.5 to 12) | 5.8 (3.9 to 7.6) | −4.7 (−7.3 to −2.1) | 0.0007 |
| Urban | 8.7 (7.3 to 10) | 0.056 (0.048 to 0.064) | 7.0 (4.5 to 9.5) | 4 (1.1 to 6.9) | −3.0 (−6.7 to 0.68) | 0.1081 |
| Quintile | | | | | | |
| Poorest | 5.2 (3.9 to 6.5) | 0.025 (0.016 to 0.033) | 12.1 (7.8 to 16) | 3.1 (1.3 to 0.049) | −9 (−14 to −4.4) | 0.0002 |
| 2nd | 0.064 (0.048 to 0.08) | 0.027 (0.021 to 0.032) | 0.095 (0.070 to 0.12) | 0.052 (0.034 to 0.070) | −0.043 (−0.073 to −0.013) | 0.0062 |
| Middle | 0.074 (0.050 to 0.098) | 0.048(0.031 to 0.065) | 0.10 (.073 to 0.133) | 0.044 (0.013 to 0.076) | −0.059 (−0.100 to −0.017) | 0.0069 |
| 4th | 0 .087(0.063 to 0.110) | 0.06 (.041 to 0.078) | 0.083 (0.061 to 0.104) | 0.039 (0.014 to 0.064) | −0.044 (−0.075 to −0.012) | 0.0076 |
| Richest | 0.10 (0.090 to 0.12) | 0.064 (0.049 to 0.079) | 0.045 (0.0035 to 0.086) | 0.049 (−0.0068 to 0.105) | 0.0045 ( −0.062 to 0.071) | 0.8937 |

**Table 6**  Changes in hospitalisation (per 1000 population) in Maharashtra and Andhra Pradesh between 2004 and 2012

| Subgroups | Baseline mean (95% CI) | | Change 2004:2012 mean (95% CI) | | DID | |
| --- | --- | --- | --- | --- | --- | --- |
| | Maharashtra | Andhra Pradesh | Maharashtra | Andhra Pradesh | Mean (95% CI) | p Value |
| Hospitalisations per 1000 population | | | | | | |
| All | 41.3 (37.3 to 45.2) | 31.5 (27.8 to 35.3) | 2.2 (−4.7 to 9.1) | 5.6 (−1.1 to 12.3) | 3.4 (−5.9 to 12.7) | 0.4636 |
| | | | | | Adjusted | |
| | | | | | 0.7 (−8.6 to 10.2) | 0.8685 |
| Head of household | | | | | | |
| Male | 41.0 (37.1 to 44.9) | 31.7 (16.9 to 34.0) | 1.9 (−5.1 to 8.8) | 4.4 (−2.4 to 11.1) | 2.5 (−6.9 to 11.9) | 0.5966 |
| Female | 51.6 (30.6 to 72.5) | 25.5 (27.7 to 35.7) | 13.9 (−7.53 to 35.4) | 41.5 (24.6 to 58.4) | 27.6 (1.1 to 54.1) | 0.0415 |
| Social group | | | | | | |
| Scheduled Tribes | 23.7 (14.2 to 33.1615) | 35.5 (21.9 to 49.1) | 17.1 (5.8 to 28.5) | −2.7 (−16.95 to 11.5) | −19.8 (−37.3 to −2.3) | 0.0272 |
| Scheduled Castes | 42.4 (36.3 to 48.4) | 29.5 (22.6 to 36.5) | 3.9 (−7.9 to 15.7) | 7.6 (−2.6 to 17.7) | 3.7 (−11.4 to 18.7) | 0.6268 |
| Other excluded | 44.2 (38.2 to 50.2) | 29.9 (25.5 to 34.3) | −1.9 (−11.2 to 7.3109) | 10.6 (3.4 to 17.8) | 12.5 (1.2 to 23.9) | 0.0309 |
| All other groups | 42.5 (37.1 to 47.9) | 34.9 (29.0 to 40.9) | 0.93 (−7.3 to 9.1) | −1.1 (−9.4 to 7.4) | −2.0 (−13.5 to 9.4) | 0.7235 |
| Location | | | | | | |
| Rural | 36.6 (32.2 to 41) | 28.9 (26.4 to 31.5) | 9.5 (3.5 to 15.4) | 8.8 (2.2 to 15.3) | −0.69 (−9.3 to 7.9) | 0.8725 |
| Urban | 48.2 (43.7 to 53) | 38.2 (35.5 to 41.2) | −7.9 (−14.5 to −1.3) | −2.5 (−12.1 to 7.1) | 5.4 (−5.8 to 16.6) | 0.3358 |
| Quintile | | | | | | |
| Poorest | 31.4 (25.9 to 36.9) | 27.5 (22.8 to 32.1) | 20.7 (9 to 32.8) | 6.4 (−1.7 to 14.4) | −14.4 (−28 to −0.31) | 0.0451 |
| 2nd | 36.5 (28.3 to 44.7) | 27.1 (21.9 to 32.4) | 8 (0.5 to 15.4) | 8.9 (−0.56 to 18.4) | 0.9 (−10 to 12.5) | 0.8746 |
| Middle | 47.8 (33.6 to 62.1) | 35.3 (29.0 to 41.6) | −1.5 (−17.9 to 14.8) | 4.1 (−0.56 to 13.7) | 5.6 (−12.8 to 24.0) | 0.5457 |
| 4th | 46.9 (39.9 to 53.9) | 41.7 (32.7 to 50.8) | −4.0 (−12 to 3.4) | −5.1 (−15.7 to 5.4) | −0.75 (−13.3 to 11.9) | 0.9056 |
| Richest | 51.0 (46 to 56.1) | 36.5 (25.8 to 47.2) | −13.1 (−18.9 to −7.1) | 1.5 (−19.5 to 22.6) | 15.0 (−5.8 to 3.6) | 0.1665 |

tribes. This is consistent with the findings of other studies,[5] [16] which reported that the slightly less vulnerable may benefit more from such schemes, with non-financial barriers undermining access to services for the most vulnerable socioeconomic groups. The case summaries below illustrate the obstacles they face. A similar observation was reported by the evaluation of the Mexican Seguro Popular health scheme which showed that a third of the treatment-cluster households who were automatically affiliated were unaware of this fact.[42]

The most likely explanation of an Aarogyasri effect may be strengthened by the changes in terms of large borrowings which have also increased over time in both states but less so in AP across all groups of analysis except the richest asset quintile. A multicountry analysis of household catastrophic health expenditure highlighted that increasing the availability of health services is critical to improving health in poor countries, but it could also raise the proportion of households facing catastrophic expenditure unless financial risk protection policies are given a high priority.[43] Borrowing of comparatively small amounts is less impactful and may be a result of improved access to financial markets and also supports consumption smoothening. During recent decades, AP has in particular witnessed a significant rise in microfinance institutions and debt due to high levels of interest levied by the more recent entrants to this market.[44] [45] Nevertheless, our results suggest that increases in large borrowings associated with inpatient healthcare were smaller in AP. Strongly significant DIDs in scheduled tribes, rural households and the poorest and second asset quintiles and moderately significant DIDs in female-headed households, scheduled castes and other excluded social groups may also point to Aarogyasri beginning to offer greater access to healthcare when families are faced with serious illness.

In terms of large OOPE, a significant DID was found only for the total households, but the direction of all DIDs except for the second asset quintile was the same as for average expenditure on inpatient care and large borrowings, that is, in favour of AP. It is perfectly possible that the non-significant differences for the majority of subgroups are due at least in part to the RSBY, the availability of private sector hospital beds for the poor and the Jeevandayee schemes reducing large OOPE in MH, and that without these schemes the expenditure would have been greater. In AP, 70% of survey households reported that they were covered by the Aarogyasri scheme, but only 25% of these 'covered' households were aware that the benefit package was limited. It is also possible therefore that in AP some families seek hospital care assuming that the Aarogyasri scheme provides comprehensive cover, but are faced with large expenditures when their treatments fall outside the limits.

The mixed picture in relation to the increasing rate of hospitalisations in both states may suggest that the Aarogyasri and, in particular, the RSBY scheme which covers common hospital procedures are addressing a large hitherto unmet need for inpatient care. The 108 scheme in AP may be an additional albeit smaller contributor. It may also be explained by a supplier-induced demand. An assessment of health provider behaviour and governance is an important strand of the evaluation of health financing schemes[46] but was outside the remit of our study. We would, however, strongly recommend an impact evaluation focusing on healthcare supply to complement our evidence.

The greater increase in hospitalisation in AP, in female-headed households and 'other excluded groups', which are two of the most vulnerable population groups, is encouraging. However, the reduction in hospitalisations among scheduled tribes and the poorest asset quintile in AP are of concern and suggest that if the poor are to secure the benefits appropriated by the near poor or more often by the rich,[47] the provision of more comprehensive health schemes is essential, which combines the tertiary and secondary care focus of Aarogyasri and the secondary care benefits of RSBY with attention paid to minimise barriers such as the widespread influence of illiteracy and lack of awareness, which limit access to even schemes such as Aarogyasri that are apparently highly inclusive and non-discriminatory, as well as distance to facilities. Our case summaries illustrate this. Furthermore, health financing reforms such as the Jamkesmas[48] in Indonesia and the Seguro Popular in Mexico,[42] both countries similar to India in terms of population and growing economies, include outpatient and inpatient care, and curative as well as preventive services, suggesting that a more comprehensive service is possible to implement and worthy of consideration.

In summary, health innovations in AP had a greater beneficial effect on hospital inpatient care-related expenditures than innovations in MH. The Aarogyasri scheme is likely to have contributed to these impacts in AP, at least in part. However, in both states, OOPE increased over time, in keeping with the picture reported nationwide.[49] The most likely explanations are that the poor spend the largest proportion of OOPE on drugs,[50] an expenditure not adequately addressed by the health financing schemes, including the RAS, which provide only 'follow-up' medicines for limited periods after discharge from hospital. Yet chronic disease such as diabetes may result in a one-off cardiovascular intervention funded by the RAS, as well as lifelong medication required to be paid for out-of-pocket. Other possible explanations are medical inflation and rising costs of initial consultation and diagnostic investigation of symptoms, often in the rapidly growing private sector outpatient services, prior to hospitalisation under the cover of the RAS or the other health financing schemes. Hospitalisations also increased, and while it is generally assumed that in developing countries this is likely to address genuine need, its impact on health status is not known.

## IMPLICATIONS FOR POLICY AND PRACTICE

Despite these uncertainties, Aarogyasri is perceived, with some justification across India, as a successful scheme, and is being rapidly replicated across the states. Since July 2012, MH too has joined the list of states offering an Aarogyasri-like scheme, the Rajiv Gandhi Jeevandayee Arogya Yojana.[51] This study has highlighted that such schemes may result in some positive outcomes. Although the study was not designed to elicit the specific features of the scheme to which any comparatively greater benefits may be attributable, and which deserve to be replicated in other states, evidence from systematic reviews[3] points to the need for schemes to be more 'comprehensively' designed to maximise their positive impact. Aarogyasri's design in terms of its aims to address financial and non-financial barriers—being fully state funded, the systematic administrative implementation of Aarogyasri across the state so that it is now almost universal, automatic enrolment, allowing access to the scheme via a ration card which most poor families own, the wide spectrum of treatments offered, the large number of healthcare providers empanelled—is perhaps responsible as a 'comprehensive' package for the greater impact.

However, the study also suggests that improved access to healthcare and the reduction of the overall burden of OOPE, especially in the most vulnerable sections of the population,[52] are likely to require additional interventions that address gaps in the availability of care and provide patients appropriate pathways that support their journey from a strong and comprehensive primary care service where they may be informed of their entitlements and investigated for their initial symptoms to appropriate hospitals for the treatment of serious illness. Besides, inpatient care is only consumed by a small proportion of households in a year,[53] and this is especially true of tertiary care, while many more will seek outpatient services and referral to inpatient care, should this be required. Others have strongly recommended the strengthening of the primary care base as an essential means to universal health coverage, and this study confirms their view.[41] The key implications for AP are to explore how best the most advantageous features of Aarogyasri can be extended to include secondary and primary care, while those for MH may be to build on and unify its menu of currently available schemes to create an evidence-based comprehensive health delivery system. These conclusions may be applicable to other states with similar health financing schemes.

The design of health financing systems as well as their evaluations is complex and challenging, as the mountain of available evidence suggests.[41 42 52 54] Even the ground-breaking Seguro Popular health insurance programme of Mexico, which used a cluster-randomised trial design with strong government support, demonstrated some but not all anticipated outcomes, contrary to expectations,[42] and a key recommendation was that continued assessment of the programme was needed.

Furthermore, differences in data and methodology may result in even very well-designed evaluations of the same programme producing contrasting findings.[52] Our evaluation is the first to use a survey methodology—the best possible in the hierarchy of evaluation methodologies, when a randomised control trial is not achievable—to evaluate the health financing reforms in two large states of India. Despite that, the study has limitations which we have acknowledged. For example, this study has examined health-related expenditure and behaviours at only two points in time. To establish trends, it is suggested that more than two time points are needed. However, it needs to be recognised that schemes such as the Aarogyasri themselves may not have longevity in their original form and may change or be replaced in response to changing needs and policy imperatives. King et al[42] acknowledged that their assessment of the Mexican Seguro Popular programme (at 10 months) was undertaken at an early stage, but it was nevertheless recognised as having provided important evidence of impacts, albeit early ones. Therefore, a realistic aim of an evaluation in this rapidly changing health delivery landscape in India would be to study change over time, even if that is limited initially to two time points, as well as to continue to evaluate the system repeatedly, and use the evidence to reshape health delivery to be responsive to future socioeconomic and epidemiological trends. In addition, the evaluation was not designed to assess provider behaviours. Many other pertinent questions, such as the impact of the schemes on the overall economy of healthcare, cannot be answered by a single evaluation. However, these are recognised problems which can only be addressed through continuous assessments, as other evaluations have shown. Our study has nevertheless produced sufficient insights to enable policy leaders to improve programme effectiveness and, importantly, to undertake further assessment. Our household survey data provide a valuable baseline for future monitoring and analyses of trends in both states.

This study needs to be followed up with further and repeated evaluations as AP's and MH's schemes evolve; to assess the impacts of redesign and to help health policy leaders achieve their aspiration of universal access to good quality healthcare.

## Three faces of the Aarogyasri scheme
### The beneficiary

Patient A, a 65-year-old widow, is from a tribal background. She is an unskilled labourer, supporting a family of 5, with a monthly income of ₹4000 (US$74). She was referred to a municipal hospital with chest pain and underwent heart surgery. The hospital, which is a part of the Aarogyasri network, provided free care and her total out-of-pocket expenses amounted to ₹850 (US$16$) for initial transport. In addition to the surgery, she received free food, money for transport home and follow-up medicines. She is very satisfied with the service she received.

## The excluded

Patient B, a mother, has a BPL card, but her daughter does not, as she was abandoned by her husband who is a government employee and entitled to free healthcare. The mother was unable to secure free care using her BPL card for her seriously ill daughter who paid out-of-pocket at a private facility where she was offered a hysterectomy for the relief of her gynaecological symptoms. Her daughter's healthcare costs were met by the family selling a number of household assets and her granddaughter discontinuing her education to take up paid work. The patient's daughter is severely depressed and does not speak to anyone.

## The uninformed

Patient C, a 43 year-old, had severe stomach pain one night. Although the family had a BPL card, her husband and son, rushed her to a private hospital nearby, which was not part of the Aarogyasri network. They were unaware of how to access the Aarogyasri scheme hospitals, which were further away from home, and the local primary health services being inadequate, patient C had not had her initial symptoms investigated. The treatment was funded through a loan from a private moneylender. On discharge, she has not attended follow-up, as the family cannot afford transport or medicines.

**Author affiliations**
[1]Institute for Health and Human Development, University of East London, London, UK
[2]Administrative Staff College of India, Hyderabad, Andhra Pradesh, India
[3]ACCESS Health International, Hyderabad, Andhra Pradesh, India
[4]SughaVazhvu Healthcare, Thanjavur, Tamil Nadu, India
[5]Indian School of Business, Hyderabad, Andhra Pradesh, India
[6]Development Research Group (DECRG), The World Bank, Washington, DC, USA
[7]Institute for Health and Human Development, University of East London & ESRC International Centre for Life Course Studies in Society and Health, University College London, London, UK

**Acknowledgements** The authors thank Bhimasankaram Pochiraju, Sundaresh Peri, C Ravi and Rahul Ahluwalia for their advice and guidance at various stages of the study design and data analysis, and to colleagues at the University of East London and Administrative Staff College of India, Hyderabad for their administrative support. The authors also thank P Suryanarayana, CM Reddy, D Chakrapani and K Pandu Ranga Reddy and AY Jadhav and his colleagues for their contribution to the training of the survey teams, verification of the survey and data collection and acknowledge the IMRB International Social Research Institute team's support in carrying out the survey. The authors are grateful to Somil Nagpal and Joseph Kutzin for commenting on the report and helping us to improve it, to Sujatha Rao for her encouragement and valuable suggestions and ideas throughout the course of the study and to Jay Bagaria for her help during the early stage of the development of the proposal. Finally, the authors thank PV Ramesh without whose constant support, guidance and encouragement this study would not have been possible.

**Contributors** MR conceived and designed the study, applied for funding, and was responsible for the supervision and management of all aspects of the study as well as the dissemination of its results. She is the guarantor. SB shared responsibility for the conception of the study, applications for funding, study design and data collation and analysis, contributed to the questionnaire design and commented on drafts of the report. PS contributed to the conception of the study and study design, led the questionnaire design and survey implementation, including training of survey staff, monitoring survey progress and data collation and verification, commented on drafts of the report and helped prepare the references. AK undertook the data collation, verification and analysis, assisted with the survey and questionnaire design and survey implementation and prepared the tables for the report. AS led the literature review, assisted with the study and questionnaire design, survey implementation and preparation and analysis of baseline data, and commented on drafts of the report. MK helped with the data analysis. AW devised the methodology for the estimation of the programme impacts, advised during the data-collection and data-preparation stages, wrote and implemented the computer code for the model estimation, helped to oversee the production of the results, and contributed text to the report. GN provided technical advice on accounting for the complex survey structure in the analysis, developed a STATA equation, helped to compute an asset index, advised on the output tables, verified the analysis and commented on drafts of the report. AR helped develop a conceptual framework for the evaluation, advised on funding proposals, the study design, analytical methodology and presentation of results and contributed text to the report. MR wrote the first draft of the paper and its redrafts in accordance with the comments of all other authors and reviewers.

**Funding** The study was funded by the International Development Research Centre, Canada, the Wellcome Trust, the UK Department for International Development, and Rockefeller Foundation. The World Bank supported Adam Wagstaff's contribution to the study.

**Competing interests** MR, AK, PS, AS and SB have support from the Rockefeller Foundation, Wellcome Trust, International Development Research Centre, Canada and Department for International Development, UK.

**Ethics approval** The study protocol and questionnaire for the 2012 survey were reviewed and approved by the Research Ethics Committee of the Administrative Staff College of India, Hyderabad, which hosted the study. Household survey questionnaires include signed consent by the head of the household or another adult representative of the household.

**Provenance and peer review** Not commissioned; externally peer reviewed.

**Data sharing statement** The 2004 National Sample Survey Organisation of India household survey questionnaire and data are available to the public. The 2012 household survey questionnaire, and the full anonymised household survey dataset will be available with open access as soon as the data analyses and submissions for publication are completed.

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
