## [Reviewer comments · BMJ Open]

Some articles will have been accepted based in part or entirely on reviews undertaken for other BMJ Group journals. These will be reproduced where possible.

ARTICLE DETAILS

TITLE (PROVISIONAL)	Changes in addressing inequalities in access to hospital care in Andhra Pradesh and Maharashtra states of India. A cross sectional difference-in-differences study
AUTHORS	Rao, Mala; Katyal, Anuradha; Singh, Prabal; Samarth, Amit; Bergkvist, Sofi; Kancharla, Manjusha; Wagstaff, Adam; Netuveli, Gopalakrishnan; Renton, Adrian

VERSION 1 - REVIEW

REVIEWER	Narayanan, Devadasan Institute of Public Health, Bangalore, India.
REVIEW RETURNED	20-Jan-2014

GENERAL COMMENTS	This is a great work and am happy that individual institutions have taken up such a study. Normally in India, we leave this to the government organisations, so congratulations to the authors for conducting such a large and ambitious study. It is also timely, because there is a lot of debate taking place about the need for evaluation of these large government sponsored health insurance interventions. I read the manuscript with interest and have the following comments. The numbering is based on the numbers of the table above. 1) In the objective, they should clearly state that they want to study the effect of Rajiv Aarogyasri rather than call it "health innovations" 2) In the abstract, the methods should clarify that they used two data sets, one before (NSSO - 2004) and other after (primary survey - 2012). From the current reading of the abstract, this is not clear. 2) In the abstract, they use Rajiv Aarogyasri only twice, but in the main manuscript, the entire description and analysis and discussion is about Rajiv Aarogyasri. So this needs to be made explicit in the abstract also. 3) If the authors were studying the effect of Rajiv Aarogyasri, then a simple case control study would have been more effective and efficient. They could have taken patients enrolled under the RA scheme and hospitalised (cases) and patients not enrolled under the RA scheme but hospitalised for similar conditions (controls) and then looked at OOPE, large spendings and borrowings in both these groups. This is especially significant RA covers mostly tertiary conditions and these are rare events. Which is ideal for a case - control study. 9) Rajiv Aarogyasri covers mostly tertiary care. This is a rare event, so to attribute the increased hospitalisation and reduced OOPE to RA is not correct. If they wanted to do this, then they should have analysed conditions that were covered under RA rather than general hospitalisation which would be for a lot of medical conditions and these are not usually covered under RA.
--

11) From the results, it appears that in AP (where RA was introduced); Hospitalisation rates are increased only for women headed households and for other excluded households. For all else categories, there is no increase in admission rate in AP. So if this is so, then naturally the OOPE and the other financial indicators would be meaningless. Because if the hospitalisation is the same or less, naturally there would be a lower OOPE and incidence of large 'expenditure' or 'borrowings'. This needs to be clearly stated in the discussion.

The authors have not dwelt much on the differences in their sampling characteristics between 2004 and 2012. While one can understand changes in urban / rural or income quintiles, there are significant differences in the caste composition between these two periods (in Maharashtra). This suggests that the two samples are not similar and could have a selection bias on the results.

There are two major gaps in this article that need to be addressed before it goes for publication.

Health expenditure is a function of the type of provider that the patient went to. For example, if the patients went mostly to government facilities, then naturally the OOPE would be low. The current results do not give us these results. As it must have been collected in the study, OOPE should be analysed as a function of the type of provider also. This is a major gap in this paper and the authors need to include this into their analysis.

Secondly, health expenditure is also determined by the type of diseases. With the RA programme in place, there would have been a higher incidence of tertiary care provided compared to Maharashtra. So the authors need to disaggregate the hospitalisations by the type of disease also to see if these have an influence on the OOPE.

These two important confounders need to be addressed before we come to any conclusion.

The mixed results (see attached file) that one finds when one analyses by sub category needs to be explained by the authors. If they do not have the explanation, then it should be acknowledged as a limitation of their study.

One minor comment - In the results section, the authors describe the results in the text as well as give the same results in the table. This is not necessary and the text can just highlight the important findings of the table. The text and table should be complementary rather than a duplicate.

Results from the study by different socio-economic characteristics of patients

	Sample	Hospitalisation	OOPE	Large expenditure	Large borrowings
Women headed HH	NS	AP > MH	NS	NS	AP < MH
Rural HH	NS	NS	AP < MH	NS	AP < MH
ST HHs	NS	MH > AP	NS	NS	AP < MH

	SC HHs	Increased significantly in Maharashtra	NS	AP < MH	NS	NS
	Other excluded HHs	Reduced significantly in Maharashtra	AP > MH	NS	NS	NS
	Q1 HHs	Reduced significantly in both states	MH > AP	AP < MH	AP < MH	AP < MH
	Q2 HHs	Reduced significantly in AP	NS	NS	NS	AP < MH

REVIEWER	V R Muraleedharan Dept of Humanities and Social Sciences Indian Institute of Technology (Madras)
REVIEW RETURNED	19-Feb-2014

GENERAL COMMENTS	Evidently, this is a useful paper. Its aim is to estimate the extent to which various policy initiatives (innovative interventions) in AP had a larger (or smaller) beneficial effects (measured in terms of HERB) than those in MH. The methodology is clear and the overall implications are well discussed. I recommend this for publication. The following points may be addressed, and some of the responses may be incorporated into the text. 1. A major point about this paper is methodological: The DID method holds good under the assumptions that “outcome determinants other than Arogyashri and RSBY are stable in both states or followed a parallel trend” (p.12) Evidently, this assumption is NOT valid. The primary purpose of any implementation research is to explain why effects of the same policy
--

/intervention across similar states (regions or districts) vary over a period of time. Given the overwhelming evidence for this observation (not only in India, but across the world), it is important to discuss the implications of this assumptions on the validity of empirical estimates arrived at using DID analysis. What happens when this assumption is dropped to the validity of the results?

The authors may add a response here on this point: Given the fact that this assumption is not valid, how well the estimates could still be interpreted from policy perspective? It is not correct to assume that OOP payments in private sector in the two states are likely to be similar in the two states. If DID method requires such unrealistic assumptions, then it's utility from policy perspective should be explained.

2. The definitions of large expenditure and large borrowings are unique to this paper. But it is not clear why the first is based on the average expenditure worked out as per the survey and the second is worked out based on the BPL threshold. No literature or reference for the same seems to be given for this.

3. From the descriptive data on the schemes it is seen that Aarogyasri covers nearly 20 million persons while RSBY covers only 2 million, and Maharashtra has a larger population than Andhra. The main finding is that the average inpatient expenditure, large expenditures and large borrowings have increased less in Andhra than Maharashtra, showing that the health innovations in Andhra are likely to have had a more beneficial effect. This can be accepted to some extent as also the statement that the Aarogyasri scheme is likely to have contributed to this beneficial effect. As observed above, the impact of RSBY in Maharashtra is definitely bound to be lower than in Andhra because the level of coverage is much lower, The last line of the ' conclusions' section is definitely true, but it is not clear how the observation on equity of access to health care is made from this study.

If data is available, it could be studied using those covered and not covered by the respective insurance schemes. Since there was no coverage under the schemes in 2004, it is possible to see the impact of the insurance schemes more clearly.

4. While studying the average inpatient cost and the other outcome parameters, it might help to look at those utilizing public and private sector for inpatient care. This would be useful for many policy purposes. The overall inpatient expenditures (not the difference) is

	consistently lower than Maharashtra, which might indicate higher use of public system, especially in rural areas. (It might be noted that the average cost of inpatient care for the poorest quintile in Maharashtra is more than the average cost for the richest quintile. ie 2342 as against 2050, whereas in Andhra, it is marginally less.) This may also contribute to the lower level of large expenditures as well as large borrowings in Andhra. What I am trying to say is, that what has been assumed as the impact of the innovative health programmes in Andhra , specifically Aarogyasri, may also be due to lower costs prima facie, which again may be due to a differential use of public and private systems, or due to lower cost of the public system itself. 5. There are many interesting observations: in the table on hospitalizations per 1000 population. However, may be they are not relevant here as the only subject of interest is the difference in difference over the two time periods between the two states. For instance the increase in hospitalization in women in Andhra is very obvious, an increase of 27.6 % with $p=0.04$ Similarly the increase in hospitalization rates of scheduled tribes in Maharashtra is 19.8% higher than Andhra.. ($p=0.02$). Last but not least, the poorest quintile in Maharashtra has shown an abnormal increase in utilization, and the richest quintile has seen a fall. Andhra has shown much less variation over the quintiles from the 2004 figures. My overall suggestion is: Points 1 and 2 above should be addressed in the relevant sections. Points 3 and 4, may be considered. Point 5 is only an observation. The paper is recommended for publication subject to these points above.
--	---

VERSION 1 – AUTHOR RESPONSE

Reviewer Name N. Devadasan

Institution and Country Institute of Public Health, Bangalore, India.

Please state any competing interests or state 'None declared': None declared

This is a great work and am happy that individual institutions have taken up such a study. Normally in India, we leave this to the government organisations, so congratulations to the authors for conducting such a large and ambitious study.

It is also timely, because there is a lot of debate taking place about the need for evaluation of these large government sponsored health insurance interventions.

Response: The authors are extremely grateful for Professor Devadasan's positive feedback

I read the manuscript with interest and have the following comments. The numbering is based on the numbers of the table above.

Is the research question or study objective clearly defined?

In the objective, they should clearly state that they want to study the effect of Rajiv Aarogyasri rather than call it "health innovations"

Response: We were erring on the side of caution, in stating that we were comparing the effects on access to health care and out of pocket expenditure of health innovations, rather than the Rajiv Aarogyasri scheme alone, because we could not exclude other innovations which had been launched contemporaneously from having contributed to any observed effects. However, we have now amended the abstract in accordance with Professor Devadasan's suggestion and stated clearly that we were interested in studying the effect of the Rajiv Aarogyasri scheme.

Is the abstract accurate, balanced and complete?

In the abstract, the methods should clarify that they used two data sets, one before (NSSO - 2004) and other after (primary survey - 2012). From the current reading of the abstract, this is not clear.

Response: We have added a methods section and clarified the use of 2 datasets. We have also revised the rest of the abstract to ensure that it complied with the word limit.

Is the abstract accurate, balanced and complete?

In the abstract, they use Rajiv Aarogyasri only twice, but in the main manuscript, the entire description and analysis and discussion is about Rajiv Aarogyasri. So this needs to be made explicit in the abstract also.

Response: We have now explicitly stated in the objectives and conclusion (the only 2 sections in the BMJ Open framework for abstracts which were appropriate for mention of this) that we were examining the effects of the Rajiv Aarogyasri scheme.

Is the study design appropriate to answer the research question?

If the authors were studying the effect of Rajiv Aarogyasri, then a simple case control study would have been more effective and efficient. They could have taken patients enrolled under the RA scheme and hospitalised (cases) and patients not enrolled under the RA scheme but hospitalised for similar conditions (controls) and then looked at OOPE, large spendings and borrowings in both these groups. This is especially significant RA covers mostly tertiary conditions and these are rare events. Which is ideal for a case - control study.

Response: We agree with the suggestion about carrying out a case control study. This is a valid and important recommendation. But we would respectfully request Professor Devadasan to consider that enrolment to the Aarogyasri scheme is automatic, and as a consequence, 85 percent of the population of Andhra Pradesh is enrolled in the scheme. We wish to refer him to line 11 in the section entitled 'The Rajiv Aarogyasri Health Insurance Scheme of AP' where we highlight this fact. Non-enrolled households were likely to be a much smaller group and of higher socio-economic status than enrolled households, and therefore would have been unsuitable as controls. In summary, the decision regarding the methodology followed extensive discussion of options including a case control study, and consultation and detailed consideration of all aspects of the scheme.

Do the results address the research question or objective?

Rajiv Aarogyasri covers mostly tertiary care. This is a rare event, so to attribute the increased hospitalisation and reduced OOPE to RA is not correct. If they wanted to do this, then they should have analysed conditions that were covered under RA rather than general hospitalisation which would be for a lot of medical conditions and these are not usually not covered under RA.

Response: The Government of Andhra Pradesh described the Aarogyasri as a scheme 'to provide treatment for serious and life-threatening illnesses with specific objectives including to improve access of poor families to quality 'tertiary' medical care (meaning low-frequency, high cost specialist care) and treatment of identified diseases requiring hospitalisation'. However, in reality, the scheme provides an

extensive range of treatments requiring secondary or tertiary care. As explained by us in lines 14-17 in the section entitled 'The Rajiv Aarogyasri Health Insurance Scheme of AP', a total of 942 medical and surgical procedures across 31 clinical specialties were provided at the time of our study. These range from for example cholecystectomies to coronary artery bypass grafts and renal dialysis. Indeed, the Aarogyasri website

(<http://www.aarogyasri.gov.in/ASRI/FrontServlet?requestType=CommonRH&actionVal=RightFrame&page=About&pageName=About&mainMenu=Home&subMenu=About>) confirms that, put together, free health care provided by the Aarogyasri scheme and by the national programmes for diseases such as AIDS, tuberculosis and malaria, have dispensed with the need for the 'below poverty line' population to approach the Government for financial assistance for individual medical problems. Professor Devadasan's comments above highlight the misunderstanding caused by the use of the word 'tertiary' in referring to the services provided by the Aarogyasri scheme. We used the word because previously, the scheme website used it. But we now find that the scheme website which included that term in its definition until 2013, no longer uses it. We thank Professor Devadasan for highlighting the uncertainty which results from the use of the word 'tertiary' to describe what is provided by the scheme, and have deleted the word from our description of the scheme in our paper. We hope that the fact that the scheme covers more than rare tertiary care events is now clear.

Are the discussion and conclusions justified by the results?

From the results, it appears that in AP (where RA was introduced); Hospitalisation rates are increased only for women headed households and for other excluded households. For all else categories, there is no increase in admission rate in AP. So if this is so, then naturally the OOPE and the other financial indicators would be meaningless. Because if the hospitalisation is the same or less, naturally there would be a lower OOPE and incidence of large 'expenditure' or 'borrowings'. This needs to be clearly stated in the discussion.

Response: Overall, hospitalisation rates showed an increase in both AP and Maharashtra but more so in AP (5.6 per 1000 population vs 2.2), although the DID was not statistically significant. Significant increases were found for female headed and 'other excluded' households, and non-significant increases for male headed and schedule caste households. We thank Professor Devadasan for highlighting the findings for female headed and 'other excluded' households, which are among the most vulnerable populations. We have revised the paper to draw particular attention to this point under the section 'Discussion', para 5, lines 8 and 9. We have also explained in lines 1-3 of para 5 in the section 'Discussion' that the non-significant DID may be due to the fact that both the Aarogyasri in AP and the RSBY in Maharashtra may be meeting some of the need for secondary care in both states. In relation to OOPE and borrowings, we respectfully request Professor Devadasan to consider that the estimates were per household. We draw his kind attention to the Section on 'Methods', Outcome measures. We believed therefore that assessing changes over time would be useful, even if rates of hospitalisation had not increased, and hope that Professor Devadasan will agree with this suggestion.

The authors have not dwelt much on the differences in their sampling characteristics between 2004 and 2012. While one can understand changes in urban / rural or income quintiles, there are significant differences in the caste composition between these two periods (in Maharashtra). This suggests that the two samples are not similar and could have a selection bias on the results.

Response: We agree that changes in socio-economic characteristics of the populations could limit the interpretation of the results. We have therefore adjusted our analysis of DID's using 9 co-variables including caste. We draw Professor Devadasan's kind attention to para 4 under the section Analysis.

There are two major gaps in this article that need to be addressed before it goes for publication.

Health expenditure is a function of the type of provider that the patient went to. For example, if the

patients went mostly to government facilities, then naturally the OOPE would be low. The current results do not give us these results. As it must have been collected in the study, OOPE should be analysed as a function of the type of provider also. This is a major gap in this paper and the authors need to include this into their analysis.

Response: We draw Professor Devadasan's kind attention to the fact that the Rajiv Aarogyasri scheme is predicated upon health expenditure no longer being a function of the type of provider which provides healthcare under the scheme. At the time of our study, 353 public and private sector hospitals were 'empanelled' to provide services to Aarogyasri beneficiaries and they were all expected to provide free care. We have explained this in the last 2 lines of para 1 in the section 'The Rajiv Aarogyasri Health Insurance Scheme of AP'. The very basis of our study was to explore the effects of such a policy (for free care, irrespective of the type of provider) on household expenditure for health care.

Secondly, health expenditure is also determined by the type of diseases. With the RA programme in place, there would have been a higher incidence of tertiary care provided compared to Maharashtra. So the authors need to disaggregate the hospitalisations by the type of disease also to see if these have an influence on the OOPE.

Response: We agree with Professor Devadasan's suggestion that expenditure may be associated with the type of disease. However, as we have pointed out in a previous section, the Aarogyasri scheme covers a very wide range of treatments for serious and life-threatening illnesses, with very few exclusions. We would therefore respectfully request Professor Devadasan to consider that the objective of our study was to compare the overall effects of health innovations over time on access to and OOPE on inpatient care in AP and MH and to assess whether the AP initiative of the Rajiv Aarogyasri scheme had larger or smaller beneficial effects than those found in MH such as the RSBY and others.

These two important confounders need to be addressed before we come to any conclusion.

The mixed results (see attached file) that one finds when one analyses by sub category needs to be explained by the authors. If they do not have the explanation, then it should be acknowledged as a limitation of their study.

Response: The description of the results which follow each table highlights the findings overall, as well as for each sub category of analysis. Furthermore, likely explanations for the findings including those for sub categories are included in considerable detail in the section 'Discussion'; average expenditure in para 2, large borrowings in para 3, large expenditures in para 4, and hospitalisations in para 5, where we have in particular offered likely explanations for the mixed picture. In the case of average expenditure, large expenditures and large borrowings, we draw Professor Devadasan's kind attention to the fact that there was a significantly larger increase in Maharashtra overall, and the general trend was in favour of AP. In terms of socio-economic changes, our regression analysis has been adjusted for covariates, and we have explained this in the section on 'Analysis', and included a further final para to strengthen our explanation. In the section on 'Limitations' we have explicitly acknowledged in para 1 that despite the adjustments, unobserved differential changes may nevertheless limit the interpretation of our findings. We would respectfully request Professor Devadasan to consider the detailed explanations included previously, together with the revisions, and the penultimate para of the section 'Implications for Policy and Practice' which once again acknowledges the limitations inherent to large and complex health financing systems and their evaluations.

One minor comment - In the results section, the authors describe the results in the text as well as give the same results in the table. This is not necessary and the text can just highlight the important findings of the table. The text and table should be complementary rather than a duplicate.

Response: We have explained the results in the text, to ensure that policy makers who may not have

the time to examine the tables nevertheless have access to the findings. As Professor Devadasan has kindly highlighted, this is an important study and highly relevant to India's policies on Government sponsored health insurance programmes. It is of great interest to policy makers, and we are keen to ensure that the write up facilitates their access to the findings. We are nevertheless happy to remove the explanations of the results in the text, should Professor Devadasan, after consideration of our response, advise us that this should be done.

Reviewer Name V R Muraleedharan

Institution and Country Dept of Humanities and Social Sciences

Indian Institute of Technology (Madras)

Please state any competing interests or state 'None declared': None Declared

Evidently, this is a useful paper. Its aim is to estimate the extent to which various policy initiatives (innovative interventions) in AP had a larger (or smaller) beneficial effects (measured in terms of HERB) than those in MH. The methodology is clear and the overall implications are well discussed.

I recommend this for publication.

Response: We are extremely grateful to Professor Muraleedharan for his unequivocal recommendation for publication.

The following points may be addressed, and some of the responses may be incorporated into the text.

1. A major point about this paper is methodological: The DID method holds good under the assumptions that "outcome determinants other than Arogyashri and RSBY are stable in both states or followed a parallel trend" (p.12)

Evidently, this assumption is NOT valid. The primary purpose of any implementation research is to explain why effects of the same policy /intervention across similar states (regions or districts) vary over a period of time. Given the overwhelming evidence for this observation (not only in India, but across the world), it is important to discuss the implications of this assumptions on the validity of empirical estimates arrived at using DID analysis. What happens when this assumption is dropped to the validity of the results?

The authors may add a response here on this point: Given the fact that this assumption is not valid, how well the estimates could still be interpreted from policy perspective? It is not correct to assume that OOP payments in private sector in the two states are likely to be similar in the two states. If DID method requires such unrealistic assumptions, then it's utility from policy perspective should be explained.

Response: We fully accept Professor Muraleedharan's point that the assumption that "outcome determinants other than Aarogyasri and RSBY are stable in both states or followed a parallel trend" is NOT valid. We mention this assumption in sentence 3 of the section entitled 'Analysis' only as the start of our explanation of our methodology, but go on to explain lower down in the same paragraph that, because such an assumption is invalid, our regression analysis was adjusted for 9 covariates including the gender of head of household, whether the household lives in a rural or urban location, the household's social group (caste), and the asset quintile to which the household belonged. Furthermore, we have included results which were unadjusted as well as adjusted for covariates in all our analysis. To make matters clearer, we have now added a new paragraph specifically addressing this issue. We draw Professor Muraleedharan's kind attention to the last para of the section 'Analysis'.

2. The definitions of large expenditure and large borrowings are unique to this paper. But it is not clear why the first is based on the average expenditure worked out as per the survey and the second is worked out based on the BPL threshold. No literature or reference for the same seems to be given for

this.

Response: We agree that our definitions of large expenditure and large borrowings are novel. Our difficulty arose from the fact that the 2004 NSSO health and morbidity survey which we were using as the baseline, included very limited data on household consumption so that an estimation of 'catastrophic health expenditure', the measure more commonly used in such studies was not possible. We have explained this under the section 'Large out-of-pocket inpatient expenditure'.

Instead, we constructed new measures of 'large' out-of-pocket expenditure and 'large' borrowings. As scientific research does permit the construction of new definitions and measures, where others are unavailable or as in this case, not possible to use due to external constraints, we took the liberty to do so. In developing our definition, we took the dictionary meaning of the word large and, following extensive discussions with stakeholders including policy leaders, decided that 'more than the average expenditure incurred by the Government of AP per Aarogyasri case' would be the threshold for large expenditure, and 'borrowing equal to or exceeding the Below Poverty Line annual earnings threshold set by Government of AP ' would be our definition of large borrowings. We have explained the basis for our definitions in the section on 'Outcome measures'. We acknowledge the relative nature of this definition. However, there are many thresholds used for even catastrophic health expenditure. And against that background, we respectfully request Professor Muraleedharan to consider our use of these novel definitions, and invite further debate and discourse on alternatives which may be used in future.

3. From the descriptive data on the schemes it is seen that Aarogyasri covers nearly 20 million persons while RSBY covers only 2 million, and Maharashtra has a larger population than Andhra. The main finding is that the average inpatient expenditure, large expenditures and large borrowings have increased less in Andhra than Maharashtra, showing that the health innovations in Andhra are likely to have had a more beneficial effect. This can be accepted to some extent as also the statement that the Aarogyasri scheme is likely to have contributed to this beneficial effect. As observed above, the impact of RSBY in Maharashtra is definitely bound to be lower than in Andhra because the level of coverage is much lower, The last line of the ' conclusions' section is definitely true, but it is not clear how the observation on equity of access to health care is made from this study.

Response: We agree with Professor Muraleedharan that levels of coverage will have affected the impacts, and the results of the study reflect the levels of commitment, leadership, and assertiveness which underpin the establishment of these schemes across the 2 states. We accept Professor Muraleedharan's point regarding equity and have replaced the word with improved access.

If data is available, it could be studied using those covered and not covered by the respective insurance schemes. Since there was no coverage under the schemes in 2004, it is possible to see the impact of the insurance schemes more clearly.

Response: Enrolment into the Aarogyasri scheme is automatic for all 'Below Poverty Line' households. and covered 85 percent of AP's population because of the State's more generous definition of BPL thresholds. We have explained this in the section 'The Rajiv Aarogyasri Health Insurance Scheme of AP'. Those not covered by the scheme would have been of higher socio-economic status in general, and unsuitable for comparison. It is for these reasons that we decided to draw comparisons between the 2 states.

4. While studying the average inpatient cost and the other outcome parameters, it might help to look at those utilizing public and private sector for inpatient care. This would be useful for many policy purposes. The overall inpatient expenditures (not the difference) is consistently lower than Maharashtra, which might indicate higher use of public system, especially in rural areas. (It might be noted that the average cost of inpatient care for the poorest quintile in Maharashtra is more than the average cost for the richest quintile. ie 2342 as against 2050, whereas in Andhra, it is marginally less.) This may also contribute to the lower level of large expenditures as well as large borrowings in

Andhra. What I am trying to say is, that what has been assumed as the impact of the innovative health programmes in Andhra , specifically Aarogyasri, may also be to due to lower costs prima facie, which again may be due to a differential use of public and private systems, or due to lower cost of the public system itself.

Response: We draw Professor Muraleedharan's kind attention to the fact that the Rajiv Aarogyasri scheme is predicated upon health expenditure no longer being a function of the type of provider which provides healthcare under the scheme. At the time of our study, 353 public and private sector hospitals were 'empanelled' to provide services to Aarogyasri beneficiaries and they were all expected to provide free care. We have explained this in the last 2 lines of para 1 in the section 'The Rajiv Aarogyasri Health Insurance Scheme of AP'. The very basis of our study was to explore the effects of such a policy (for free care, irrespective of the type of provider) on household expenditure for health care. The data show that admissions to private hospitals have increased over time in both states, and we do plan to carry out further analysis on this aspect of change. We thank Professor Muraleedharan for his valuable suggestion for further analysis which we will undertake for our next paper.

5. There are many interesting observations: in the table on hospitalizations per 1000 population. However, may be they are not relevant here as the only subject of interest is the difference in difference over the two time periods between the two states. For instance the increase in hospitalization in women in Andhra is very obvious, an increase of 27.6 % with $p=0.04$ Similarly the increase in hospitalization rates of scheduled tribes in Maharashtra is 19.8% higher than Andhra.. ($p=0.02$). Last but not least, the poorest quintile in Maharashtra has shown an abnormal increase in utilization, and the richest quintile has seen a fall. Andhra has shown much less variation over the quintiles from the 2004 figures.

Response: We thank Professor Muraleedharan for his valuable observations which we will use to plan further analysis of our data.

My overall suggestion is: Points 1 and 2 above should be addressed in the relevant sections. Points 3 and 4, may be considered. Point 5 is only an observation. The paper is recommended for publication subject to these points above.

Additional response: We request our reviewers to kindly note that the figure (a map showing the locations of the 2 states) from a previous version of the paper has now been deleted, as it is not of adequate resolution to be published.

VERSION 2 – REVIEW

REVIEWER	Narayanan, Devadasan Institute of Public Health Bangalore India
REVIEW RETURNED	29-Apr-2014

GENERAL COMMENTS	The authors have addressed most of the comments that I have raised in my earlier review. However, in the process, they raise an important question that has not been answered. They say that 85% of the people in AP are covered by the RA scheme. And that RA scheme covers most of the secondary as well as tertiary conditions. In which case, why has the OOP expenses increased in 2012. I would have expected a drastic fall in OOP expenses. Would appreciate if the authors could explain the reason for this. I would request the authors to address my question in their
--

	discussion. And if the editors find it satisfactory, they may accept the paper for publication.
--	---

VERSION 2 – AUTHOR RESPONSE

We thank Dr Devadasan for approving all our explanations to points he had raised previously. In response to his point raised above, we wish to draw attention to the studies which have consistently shown that OOP expenses have risen nationwide over time, and that there are a number of likely explanations which illustrate that health financing schemes such as the RAS which cover only hospitalisation are not adequate to reduce OOPE. We have inserted lines 3 - 11 in the final para of discussion to explain why the rise of OOPE despite the launch of the schemes and the extensive population eligibility of the RAS. These explanations for rising OOPE link appropriately with para 2 of the section 'Implications for policy and practice' where we have suggested that schemes such as the RAS alone are unlikely to reduce OOPE, and recommended that it needs to be part of a 'whole system' of health delivery.